# Can We Go beyond Pathology? The Prognostic Role of Risk Scoring Tools for Cancer-Specific Survival of Patients with Bladder Cancer Undergoing Radical Cystectomy

**DOI:** 10.3390/biomedicines12071541

**Published:** 2024-07-11

**Authors:** Aleksander Ślusarczyk, Rafał Wolański, Jerzy Miłow, Hanna Piekarczyk, Piotr Lipiński, Piotr Zapała, Grzegorz Niemczyk, Paweł Kurzyna, Andrzej Wróbel, Waldemar Różański, Piotr Radziszewski, Łukasz Zapała

**Affiliations:** 1Department of General, Oncological and Functional Urology, Medical University of Warsaw, 02-091 Warsaw, Poland; 22nd Clinic of Urology, Medical University of Lodz, 93-513 Łódź, Poland; 3Second Department of Gynecology, Medical University of Lublin, Jaczewskiego 8, 20-090 Lublin, Poland

**Keywords:** radical cystectomy, bladder cancer, comorbidities, Charlson comorbidity index, cancer-specific survival, AJCC system

## Abstract

Radical cystectomy (RC) remains a mainstay surgical treatment for non-metastatic muscle-invasive and BCG-unresponsive bladder cancer. Various perioperative scoring tools assess comorbidity burden, complication risks, and cancer-specific mortality (CSM) risk. We investigated the prognostic value of these scores in patients who underwent RC between 2015 and 2021. Cox proportional hazards were used in survival analyses. Risk models’ accuracy was assessed with the concordance index (C-index) and area under the curve. Among 215 included RC patients, 63 (29.3%) died, including 53 (24.7%) cancer-specific deaths, with a median follow-up of 39 months. The AJCC system, COBRA score, and Charlson comorbidity index (CCI) predicted CSM with low accuracy (C-index: 0.66, 0.65; 0.59, respectively). Multivariable Cox regression identified the AJCC system and CCI > 5 as significant CSM predictors. Additional factors included the extent of lymph node dissection, histology, smoking, presence of concomitant CIS, and neutrophil-to-lymphocyte ratio, and model accuracy was high (C-index: 0.80). The internal validation of the model with bootstrap samples revealed its slight optimism of 0.06. In conclusion, the accuracy of the AJCC staging system in the prediction of CSM is low and can be improved with the inclusion of other pathological data, CCI, smoking history and inflammatory indices.

## 1. Introduction

Radical cystectomy (RC) remains a mainstay surgical treatment for non-metastatic muscle-invasive (MIBC) and BCG-unresponsive non-muscle invasive bladder cancer (NMIBC) [1]. RC, while an essential intervention for muscle-invasive bladder cancer, is frequently associated with elevated perioperative complication rates and diminished quality of life outcomes [2,3]. Nevertheless, while these complications might be considered acceptable if the oncological results were unequivocal, MIBC is known for its aggressive course and substantial risk of lymph node involvement and distant metastasis, with a 5-year overall survival (OS) rate not exceeding 60% [1,4]. Furthermore, long-term cystectomy survivors may experience cancer in the remnant urothelium (e.g., upper tract urothelial cancer) and complications related to urinary diversions [4,5].

Various risk-scoring tools are employed in the perioperative setting to gauge the burden of comorbidities, complication risks, and the risk of cancer-specific death [1,6,7,8]. These tools underpin the ‘pentafecta’ concept, a novel approach assessing the quality of radical cystectomy [9]. Among them, user-friendly clinical tools like the Charlson comorbidity index (CCI) are commonly employed to evaluate the burden of concomitant diseases and perioperative risks [6]. Recently, the Cancer of the Bladder Risk Assessment (COBRA) score was introduced to provide a more comprehensive prognosis consultation for patients after RC, incorporating factors such as age at the time of surgery, tumor T stage, and lymph node density [7]. Other pivotal considerations in the perioperative assessment of radical cystectomy candidates include risk calculations for venous thromboembolism, with the Caprini score being one of the most widely recognized [8]. The imperative to establish robust guidelines for clinical risk assessment is evident in the quest to enhance curative effects and reduce complication rates. Our hypothesis posits that a shift toward personalized management could hold the key to improved outcomes for MIBC patients undergoing RC.

The intensification of oncological care in the perioperative setting encompasses perioperative chemotherapy, adjuvant immunotherapy, more extensive lymph node dissection (LND), and potentially radiation therapy in select cases [1,10]. Remaining at the forefront of advancements, we observe the evolution of immunotherapy and antibody-drug conjugates as viable options in the metastatic setting, anticipating their expanded use in less advanced stages pending clinical trial outcomes [11,12]. Conversely, a less intensive oncological treatment approach may be suitable for patients with a lower anticipated cancer-specific mortality. This approach involves preserving neurovascular bundles, sparing female sexual organs, and employing refined techniques for urinary diversion [1,13]. The refinement of perioperative risk assessment tools holds the promise of guiding optimal, risk-based decisions regarding surgical approaches, the extent of surgery, the choice of urinary diversion, and the administration of perioperative systemic therapies. By transcending conventional reliance on pathological data, there is an opportunity to fortify the efficacy of existing prognostic tools.

We aimed to evaluate the prognostic role of different perioperative risk scores for cancer-specific mortality in patients with bladder cancer undergoing radical cystectomy.

## 2. Materials and Methods

### 2.1. Study Design

This multicenter retrospective study included patients who underwent RC in two academic centers between 2015 and 2021. We included consecutive patients with bladder cancer staged pTis-T1–T4 N0–3 M0 who underwent RC. We excluded patients with missing follow-up data (*n* = 5), distant metastasis at the time of RC (*n* = 16), and salvage palliative cystectomy due to symptoms (*n* = 26). The flowchart showing selection criteria is presented in Figure 1.

All patients underwent RC with pelvic lymphadenectomy and urinary diversion. The decision about the type of urinary diversion was made considering patient and disease-associated factors or surgeon preference. Administration of neoadjuvant chemotherapy was performed in selected patients after oncological consultations and at the treating physicians’ discretion.

### 2.2. Data Acquisition

Systematic preoperative tests included the following: routine laboratory tests, medical history, comorbidity assessment, mortality and thromboembolism risk assessment with validated questionaries such as the Charlson comorbidity index and Caprini score [6,8]. The 8th American Joint Committee on Cancer (AJCC), TNM system, and the COBRA and simplified COBRA risk scores were calculated as previously [7,13,14].

Contrast-enhanced computed tomography imaging of the chest, abdomen and pelvis, and pathological report from the transurethral resection of the bladder tumor were consecutively used for preoperative clinical staging.

The following data were collected: gender, age, American Association of Anesthesiology (ASA) scale, body mass index (BMI), smoking status, neoadjuvant systemic treatment, pre- and postoperative TNM staging, surgical modality, type of urinary diversion, the incidence of positive surgical margin, number of resected lymph nodes (LN), length of hospitalization, postoperative course, and the occurrence of complications classified according to the Clavien–Dindo scale. All data were obtained from the medical charts and databases from the enrolled centers. Additionally, telemedicine visits were conducted considering incomplete follow-up details, whenever necessary.

Inflammatory indices such as neutrophil to lymphocyte ratio (NLR), neutrophil to erythrocyte ratio (NER), and systemic immune-inflammation index (SII—neutrophil × platelet/lymphocyte) were calculated as previously [15].

Survival time was calculated from the date of RC. Patients were censored at the last follow-up. Cancer-specific mortality (CSM) was defined as death related to cancer progression or cancer-related treatment. Overall mortality (OM) was defined as death to any cause in the follow-up.

### 2.3. Ethics Statement

Due to the character of the study, the Institutional Review Board waived the need for study approval. The study was performed in accordance with the Declaration of Helsinki and its later amendments.

### 2.4. Statistical Analysis

The number of patients and percentage for categorical variables and medians, accompanied by the interquartile range (IQR) for continuous variables, are presented in respective tables. Fisher’s test or Chi-square test for categorized variables and the Mann–Whitney U test for continuous variables were used to determine the differences between groups.

The Kaplan–Meier method and Cox proportional hazards were used for survival analyses. Survival estimates were derived from Kaplan–Meier curves. The log-rank test was used for the comparison of Kaplan–Meier curves. Median survival was computed with the reverse Kaplan–Meier method.

The external validation of available models and associations between variables and survival were tested with Cox proportional hazards. The optimal cut-off values for the inflammatory markers or risk scores were determined using the receiver operating characteristics curve. Univariable analyses utilized Cox proportional hazards to identify risk factors for CSM and OM. Significant variables in the univariate analyses were included in subsequent multivariable analysis with the stepwise selection of variables. Cox regression was employed to derive hazard ratios (HR) accompanied by 95% confidence intervals (95% CI). The accuracy of risk models was assessed with the concordance index (C-index) and area under the curve (AUC).

To internally assess the effectiveness of our risk model and address the risk of overfitting, we utilized a bootstrap resampling approach. This involved creating 300 bootstrap samples by randomly drawing with replacements from the original dataset while retaining the initial sample size. We then calculated optimism for the C-index.

A two-sided *p*-value of <0.05 was deemed statistically significant for all conducted statistical analyses. All statistical analyses were performed in SAS software version 9.4 (SAS Institute, Cary, NC, USA).

## 3. Results

Overall, 215 patients who underwent RC were included; 186 for MIBC and 29 for high- and very-high-risk NMIBC. The cohort comprised 152 males and 63 females. The median patient’s age was 68 years (IQR 64–73). All patients were of white race and Central-Eastern European origins and were treated in the academic institution. The vast majority of patients were active or former smokers (178; 83%) and only 26 (12%) denied smoking. Median BMI was 25.3 kg/m^2^ with an interquartile range between 22.2 and 29.3 kg/m^2^. At the median follow-up of 39 months (IQR 21–60 months), 63 patients (29.3%) died due to all causes, including 53 (24.7%) cancer-specific deaths.

We included patients with bladder cancer distributed across different pathological T stages, as follows: pT0 (7.91%), pT1 (8.84%), pT2 (24.19%), pT3 (30.70%), pT4 (18.14%), pTa (2.33%), and CIS (7.91%). Post-RC pathology revealed downstaging to <pT2 in the proportion of initial MIBC tumors in patients who received neoadjuvant chemotherapy (64; 30%). Nodal staging had the following distribution: pN0 (72.56%), pN1 (14.88%), pN2 (11.16%), and pN3 (1.4%). The majority of surgeries were conducted using an open approach (208; 96.7%).

Risk stratification with the AJCC system demonstrated the following distribution: stage 0 in 38 patients (17.67%), stage I in 17 patients (7.91%), stage II in 40 patients (18.60%), stage IIIa in 92 patients (42.79%), and stage IIIb in 27 patients (12.56%).

The median COBRA score was 2 (IQR 1–3). Simplified COBRA stratified 95 (44.2%), 104 (48.4%) and 16 (7.4%) patients into low-, intermediate-, and high-risk groups, respectively. Baseline characteristics regarding clinicopathological factors and risk-scoring tools were presented in Table 1 and Table 2, respectively. Estimates of 2- and 5-year cancer-specific survival (CSS) were 84.7% and 63.1%. Estimates of 2- and 5-year OS were 84.3% and 57%.

### 3.1. Validation of Risk Models

External validation of the AJCC system and COBRA score confirmed their predictive value for CSM, although their accuracy was low (C-index/5-year AUC of 0.66/0.72 and 0.65/0.70, respectively). The validation of morbidity risk scores demonstrated an association between the Charlson comorbidity index and cancer-specific mortality (C-index/5-year AUC of 0.60/0.58), whereas the Caprini score showed no significant association (C-index/5-year AUC of 0.50/0.56). Of note, the association of non-age adjusted CCI and CSM was at the border of statistical significance (C-index 0.58 and *p* = 0.055), whereas the ASA scale was not associated with CSM (C-index 0.52 and *p* = 0.55). Kaplan–Meier curves illustrated CSS, according to the AJCC system, simplified COBRA risk groups, CCI value, ASA scale, Caprini score, and NLR value (Figure 2). CSS was significantly worse in patients with higher AJCC stage, higher COBRA score, and higher NLR (all *p* < 0.05).

### 3.2. Univariable Analyses for Cancer-Specific Mortality

Univariable analyses with Cox proportional hazards revealed the following risk factors for CSM: pN stage (*p* < 0.01), lymph node density (*p* < 0.01), lymph node yield (*p* = 0.1), concomitant CIS (*p* = 0.12), histology type (*p* < 0.05), surgical margin status (*p* < 0.05), positive urethral margins (*p* < 0.05), active smoking (*p* < 0.05), blood transfusion (*p* < 0.05), length of hospital stay (*p* < 0.05), AJCC system (*p* < 0.05), COBRA nomogram (*p* < 0.01), simplified COBRA risk groups (*p* < 0.01), Charlson comorbidity index (*p* < 0.05), non-age adjusted Charlson comorbidity index (*p* = 0.05), neutrophil to lymphocyte ratio (*p* < 0.05), systemic immune-inflammation index (*p* < 0.05), and neutrophil to erythrocyte ratio (*p* < 0.05) (Table 3).

### 3.3. Multivariable Analysis for Cancer-Specific Mortality

We constructed a multivariable Cox regression model, including risk scores, and found that the AJCC system (II vs. 0: HR = 2.59, 95% CI = 0.62–10.74, *p* = 0.19; IIIa vs. 0: HR = 3.39, 95% CI = 1.01–11.39, *p* = 0.05; IIIb vs. 0: HR = 23.25, 95% CI = 6.06–89.3, *p* < 0.01) and high Charlson comorbidity index (CCI > 5: HR = 2.72, 95% CI = 1.45–5.11, *p* < 0.01) were important predictors of CSM. Other significant factors that increased the accuracy of the AJCC-based model were the number of dissected LNs (HR = 0.91, 95% CI = 0.86–0.97, *p* < 0.01), histology type (other vs. urothelial: HR = 21.24, 95% CI = 3.97–114, *p* < 0.01; SCC vs. urothelial: HR = 2.31, 95% CI = 0.27–20.18, *p* = 0.45), smoking status (never vs. former: HR = 2.71, 95% CI = 1.06–6.9, *p* = 0.04; active vs. former: HR = 2.85, 95% CI = 1.47–5.53, *p* < 0.01), concomitant CIS (HR = 18.09, 95% CI = 3.41–96.03, *p* < 0.01), and high neutrophil-to-lymphocyte ratio (HR = 1.80, 95% CI = 1.01–3.21, *p* = 0.046) as independent predictors for CSM (Table 4 and Figure 3).

Our CSM model exhibited a C-index of 0.80 and the AUCs for the 2-year and 5-year prediction of CSM were 0.84 and 0.85. In the bootstrapped cohort comprising 300 resampled cohorts, the model underwent internal validation, revealing an optimism in the C-index of 0.06.

### 3.4. Multivariable Analysis for Overall Mortality

Similar risk factors were identified as prognostic for overall mortality (Table 5). Multivariable Cox proportional hazards showed that AJCC (II vs. 0: HR = 1.79, 95% CI = 0.49–6.56, *p* = 0.38; IIIa vs. 0: HR = 3.02, 95% CI = 1.06–8.63, *p* = 0.04; IIIb vs. 0: HR = 18.34, 95% CI = 5.47–61.51, *p* < 0.01), number of dissected LNs (HR = 0.92, 95% CI = 0.87–0.96, *p* < 0.01), histology type (other vs. urothelial: HR = 10.84, 95% CI = 2.13–55.13, *p* = 0.01; SCC vs. urothelial: HR = 2.31, 95% CI = 0.26–20.94, *p* = 0.46), concomitant CIS (HR = 9.49, 95% CI = 1.92–46.94, *p* = 0.01), high Charlson comorbidity index (CCI > 5: HR = 1.82, 95% CI = 1–3.3, *p* = 0.05), age (HR = 1.04, 95% CI = 1–1.08, *p* = 0.09), smoking (never vs. former: HR = 2.0, 95% CI = 0.9–4.7, *p* = 0.10; active vs. former: HR = 2.55, 95% CI = 1.36–4.78, *p* < 0.01), and the high systemic immune inflammation index (>650: HR = 1.99, 95% CI = 1.03–3.87, *p* = 0.04) were significant predictors of overall mortality.

Our OM model exhibited a C-index of 0.77 and the AUCs for 2-year and 5-year prediction of OM were 0.83 and 0.81. In the bootstrapped cohort comprising 300 resampled cohorts, the model underwent internal validation, revealing an optimism in the C-index of 0.059.

## 4. Discussion

In our retrospective bi-center study, we evaluated the accuracy of different risk scores in the prediction of CSM following radical cystectomy for bladder cancer in search of greater personalization of the available clinical risk assessment tools.

The accuracy of the AJCC staging system in the prediction of CSM is low and could be improved with the inclusion of other pathological data, Charlson comorbidity index, smoking history, and inflammatory indices. Additional pathological data that should be incorporated into risk models encompasses the number of removed LNs, the presence of concomitant CIS, and histology type, with non-urothelial histology as a risk factor for adverse outcomes.

Furthermore, we confirmed that the burden of comorbidities summarized with CCI is prognostic, not only for OM but also for CSM. On the other hand, the thromboembolism risk score, such as the Caprini index, were not predictive for CSM nor for OM. It is worth noting that age is an important component of above-risk scores, but appeared to be not associated with CSM when a conventional level of statistical significance was applied. Non-age-adjusted CCI was also predictive for CSM, suggesting that non-age-dependent comorbidity burden impairs the prognosis. Finally, the indicators of inflammatory state such as NLR and habits such as active smoking might be additional factors considered in risk assessment. Smoking was the only potentially modifiable patient-dependent factor that could influence survival. The complementary role of NLR reflects the importance of immune capability in the outcomes of oncological surgery.

The rationale for this study comes from the need for a better understanding of the aggressiveness of advanced urothelial carcinoma which, despite great advances in oncological and urological care, is associated with poor prognosis. Many studies evidenced that within the last 20 years, the survival following RC did not improve significantly [16]. This situation could have been attributed to the biological aggressiveness of the disease, but also to the lack of effective systemic therapy or surgical inadequacy. Surgical waiting lists are another important factor known to affect survival due to very rapid disease progression, implying a short window of opportunity for surgery with curative intent [17,18]. No better treatment than RC has been invented for MIBC nor for BCG-unresponsive NMIBC despite the emergence of bladder-sparing approaches with trimodal therapy including radiation, chemotherapy, and maximal transurethral resection. However, some studies even suggested the equivocal outcomes of RC and TMT, but confounders and selection bias are the major limitations of these reports [19].

In this context, our study provides up-to-date outcomes of RC that are not satisfactory and identifies limitations of current scoring systems and risk tools. We believe that AJCC, COBRA, and the Charlson comorbidity index should be routinely used to risk-stratify patients and select candidates for adjuvant therapies (e.g., nivolumab/ radiation/ platin-based chemotherapy), but at the same time, we should seek for new biomarkers to personalize treatment. Most promising biomarkers include cell-free tumor DNA, which has been elegantly shown to predict the benefits of adjuvant therapy with atezolizumab [20]. Meanwhile, our risk model provides easy-to-use prognostic factors that can be listed during the postoperative counseling of patients to offer adjuvant therapy and present the risk.

Available evidence from the systematic review comparing different tools to assess the comorbidity burden in the context of BC patients undergoing RC indicates that CCI is the most suitable one [21]. On the other hand, the results of different studies were conflicting regarding the role of CCI in the prediction of CSM [21,22,23]. There is an agreement that CCI is a good predictor of 90-day perioperative mortality and overall mortality [21,22]. A study by Mayr et al. has shown that CCI is prognostic for cancer-independent mortality, but not CSM, and therefore it is a reliable tool supporting competing risk assessment [22]. However, three other studies reported that CCI is an independent predictor of CSM [23,24,25]. We might suspect that the reason for the association of comorbidities with CSM may underlie in the compromised immune function, the de-escalation of treatment due to reduced tolerance, and the interference of comorbid conditions with treatment response.

Our results showed that the AJCC system is equivalent to COBRA in the prediction of CSM, considering their accuracy. We showed that the discrimination abilities of COBRA nomogram reflected with a C-index of 0.65 correspond with the results of its internal validation in the development population (C-index of 0.68–0.705) [7]. On the other hand, a previous large retrospective study which validated COBRA has shown its disadvantage when compared to the AJCC system in terms of CSM prediction accuracy [14]. Muilwijk et al. suggested the simplification of COBRA but it did not lead to its superiority over the AJCC system [14]. Our validation showed that simplified COBRA properly stratifies according to CSS.

The risk of thromboembolism is always considered before RC and prophylaxis is indicated, including compression socks and low-molecular-weight heparin. The Caprini score is the most recognized tool to assess the risk of such events and guides the extent of prophylaxis. In the large cohort of 2316 patients treated with open RC, a rate of 4.7% of symptomatic thromboembolic events was reported [26]. Of these cases, 57.8% developed VTE after discharge at home, at a median of 20 days postoperatively [26]. In the contemporary cohort of patients undergoing robotic RC, Elsayed reported that 5% of patients experience venous thromboembolism, which is similar to previous results following the open approach [27]. Another study compared the risks of VTE among different major urological surgeries and demonstrated that RC was associated with increased risk which was reflected by 5% of VTE events following RC [28]. In our study, thromboembolism risk assessed with the Caprini score was not associated with OM or CSM.

Therefore, the comorbidity burden but not thromboembolism risk factors might be considered in the survival prognosis. Our multivariable model including seven risk factors revealed high accuracy, with evidence of slight overfitting verified in the bootstrap validation cohorts. The estimation of CSM should rely on four groups of factors including pathological data (AJCC system, concomitant CIS, histology type), a reflection of surgical quality (LN count), comorbidity burden (age-adjusted CCI), and patient behavior (smoking cessation and immune competence/inflammatory response). LND is of therapeutic and prognostic significance in urothelial cancer and established evidence for better survival with a higher number of resected LNs [29,30,31]. LND might reflect the quality of surgery and is included as a pentafecta criterion with different cut-offs for the number of LNs resected [3,9]. Smoking is a well-known risk factor for adverse outcomes in NMIBC and MIBC patients, increasing recurrence risk, impeding response to neoadjuvant chemotherapy, and worsening CSM following RC [32,33]. Notably, our findings indicate that active smokers are associated with a worse prognosis, while former smokers demonstrate a better prognosis compared to never-smokers. A previous study has demonstrated that quitting smoking more than 10 years before RC mitigates the risk of recurrence and CSM following the procedure [34,35]. Furthermore, a lack of balance between immune competence and inflammatory response reflected by indices such as neutrophil-to-lymphocyte ratio is a well-recognized risk factor for adverse oncological outcomes following different urological surgeries, including RC [36,37]. A meta-analysis of eighteen studies demonstrated that higher NLR was associated with worse recurrence-free and overall survival [38]. Non-urothelial histology was shown as a risk factor associated with advanced stage and higher mortality [39]. Concomitant CIS was considered a proxy of disease aggressiveness and portended worse survival outcomes [40].

Limitations of our study are inherent due to its retrospective nature. Due to a relatively short follow-up, we were not able to validate the risk models for the prediction of long-term mortality outcomes. The results of our study and validation analysis must be contextualized within the relatively homogeneous Central-Eastern European origins of the included patients. A relatively low percentage of patients received neoadjuvant chemotherapy. Neoadjuvant chemotherapy could have led to downstaging in responders, and the inclusion of neoadjuvant chemotherapy status introduces a confounding factor in the analysis of pT stage. Our risk models were internally validated in the 300 bootstrapped cohorts and revealed slight overfitting reflected by the C-index optimism of 0.06.

## 5. Conclusions

The accuracy of the AJCC staging system and COBRA risk score is low in the prediction of CSM among the contemporary cohort of bladder cancer patients undergoing radical cystectomy. The predictive value of the AJCC system can be improved with the inclusion of other pathological data (histology and concomitant CIS), lymph node counts, the Charlson comorbidity index, smoking history, and systemic inflammatory markers. The further incorporation of molecular markers and cell-free tumor DNA is expected to facilitate better risk stratification.

## Figures and Tables

**Figure 1 biomedicines-12-01541-f001:**
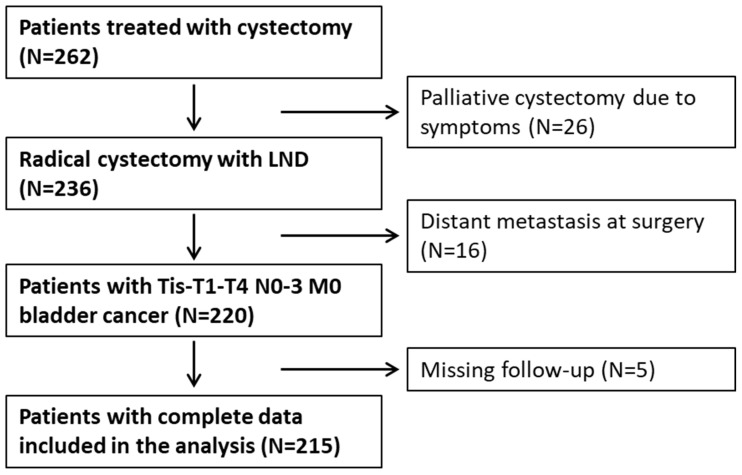
Flowchart illustrating selection criteria.

**Figure 2 biomedicines-12-01541-f002:**
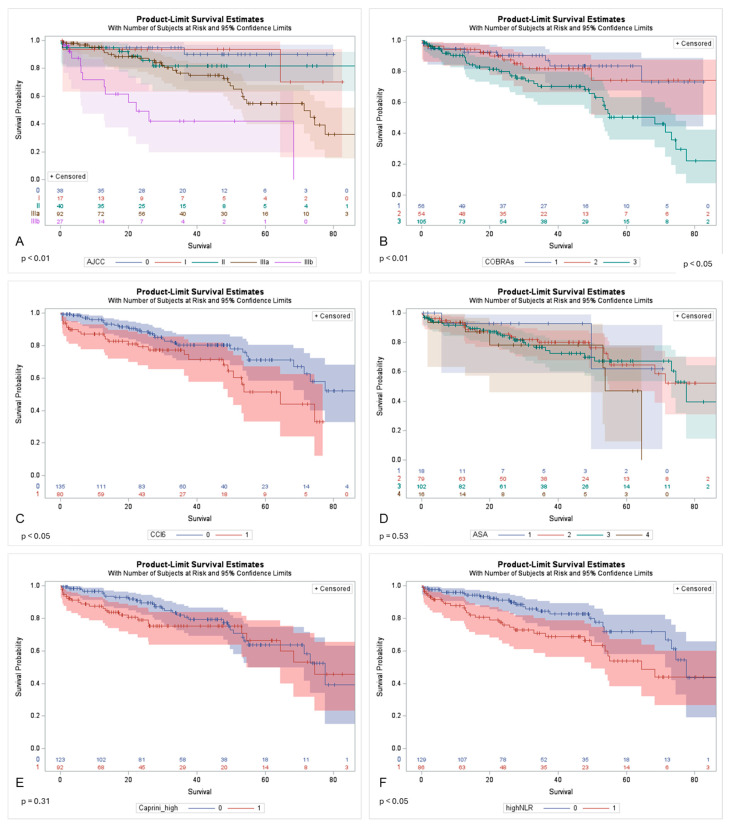
Kaplan–Meier curves with confidence intervals illustrating cancer-specific survival (months) according to (**A**) AJCC system, (**B**) simplified COBRA stratification, (**C**) Charlson comorbidity index (CCI >5 vs. ≤5), (**D**) ASA scale, (**E**) Caprini score (>6 vs. ≤6), and (**F**) neutrophil to lymphocyte ratio (NLR) (>3.5 vs. ≤3.5). Numbers at risk are presented below the curves at specific time points during follow-up measured within months following RC.

**Figure 3 biomedicines-12-01541-f003:**
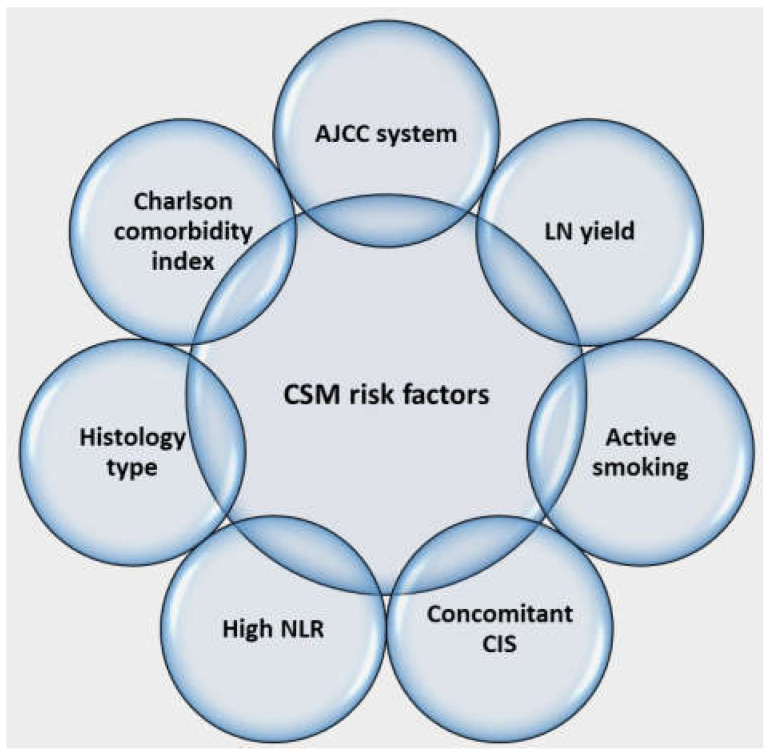
Graphical visualization of a multivariable risk model for CSM.

**Table 1 biomedicines-12-01541-t001:** Baseline characteristics of patients with bladder cancer who underwent radical cystectomy.

Variables		Whole Cohort	
		No. of Pts/Median	% of Patients/IQR
Gender	male	152	70.7
	female	63	29.3
Age	years	68	64–73
BMI	kg/m^2^	25.3	22.2–29.3
pT stage	0	17	7.91
	1	19	8.84
	2	52	24.19
	3	66	30.70
	4	39	18.14
	Ta	5	2.33
	CIS	17	7.91
Tumor grade	HG	204	94.88
	LG	11	5.12
Maximal staging	NMIBC	29	13.49
	MIBC	186	86.51
pN stage	0	156	72.56
	1	32	14.88
	2	24	11.16
	3	3	1.4
Lymph node density	0	156	72.56
	0–0.33	48	22.33
	0.34–0.49	4	1.86
	≥0.50	7	3.26
Lymph node yield	number	10	6–15
Concomitant CIS	no	210	97.67
	yes	5	2.33
Histology	TCC	206	95.81
	SCC	6	2.79
	other	3	1.40
Surgical margin status	R0	180	83.72
	R1	30	13.95
	R2	5	2.33
Ureteral margin	negative	201	93.49
	positive	14	6.51
Urethral margin	negative	202	93.95
	positive	13	6.05
Length of hospitalization	days	15	10–19
Smoking status	never	26	12.09
	former	107	49.77
	active	71	33.02
	unknown	11	5.12
Blood transfusion	no	95	44.19
	yes	120	55.81
Neoadjuvant chemotherapy	no	151	70.23
	yes	64	29.77

BMI—body mass index; CIS—carcinoma in situ; HG—high-grade; LG—low-grade; MIBC—muscle-invasive bladder cancer; NMIBC—non-muscle-invasive bladder cancer, TCC—transitional cell carcinoma; SCC—squamous cell carcinoma.

**Table 2 biomedicines-12-01541-t002:** Risk scoring models and immune-inflammatory markers in patients with bladder cancer who underwent radical cystectomy.

Variables		Whole Cohort	
		No. of Pts/Median	% of Patients/IQR
AJCC system	0	38	17.67
	I	17	7.91
	II	40	18.60
	IIIa	92	42.79
	IIIb	27	12.56
COBRA nomogram	0	53	24.65
	1	42	19.53
	2	15	6.98
	3	55	25.58
	4	34	15.81
	5	9	4.19
	6	7	3.26
Simplified COBRA	low risk	95	44.19
	intermediate risk	104	48.37
	high risk	16	7.44
ASA scale	1	18	8.37
	2	79	36.74
	3	102	47.44
	4	16	7.44
Charlson comorbidity index	score	5	4–6
Charlson comorbidity index	≤5	135	62.79
	>5	80	37.21
Caprini score	≤12	166	77.21
	>12	49	22.79
Improve score	≤3	180	83.72
	>3	35	16.28
Neutrophil to lymphocyte ratio	≤3.5	86	40
	>3.5	129	60
Systemic immune-inflammation index	≤650	76	35.35
	>650	139	64.65
Neutrophil to erythrocyte ratio	≤1.2	81	37.67
	>1.2	134	62.33
Hemoglobin concentration	<12 g/dL	121	56.28
	≥12 g/dL	94	43.72

AJCC—American Joint Committee on Cancer; COBRA—Cancer of the Bladder Risk Assessment; ASA—American Society of Anesthesiology.

**Table 3 biomedicines-12-01541-t003:** Univariable analysis with Cox proportional hazard for cancer-specific mortality among patients with bladder cancer who underwent radical cystectomy.

Variables		HR	95% CI	*p*-Value
Gender	female	ref		
	male	1.14	0.61–2.13	0.69
Age	years	1.03	0.99–1.06	0.17
BMI	kg/m^2^	0.95	0.88–1.03	0.21
pT Stage	0	ref		
	1	2.01	0.37–10.96	0.42
	2	1.56	0.34–7.23	0.57
	3	2.57	0.59–11.15	0.21
	4	5.07	1.18–21.86	0.03
	Ta	1.47	0.13–16.3	0.75
	CIS	0.53	0.05–5.8	0.6
Tumor grade	LG	ref		
	HG	3.09	0.43–22.37	0.26
pN stage	0	ref		
	1	1.77	0.86–3.62	0.12
	2	5.83	2.85–11.95	<0.01
	3	2.29	0.31–16.91	0.42
Lymph node density	0	ref		
	0–0.33	2.31	1.27–4.2	0.01
	0.34–0.49	5.83	0.76–44.67	0.09
	≥0.50	8.17	2.81–23.79	<0.01
Lymph node yield	number	0.96	0.92–1.01	0.1
Concomitant CIS	no	ref		
	yes	3.14	0.75–13.15	0.12
Histology	TCC	ref		
	other	5.7	1.36–23.93	0.02
	SCC	0.71	0.1–5.16	0.73
Surgical margin status	R0	ref		
	R1	2.19	1.14–4.21	0.02
	R2	13.89	2.91–66.37	<0.01
Ureteral margin	negative	ref		
	positive	1.71	0.77–3.83	0.19
Urethral margin	negative	ref		
	positive	2.45	1.1–5.44	0.03
Length of hospitalization	days	1.02	1–1.03	0.03
Smoking status	former	ref		
	never	1.89	0.81–4.42	0.14
	active	2.61	1.41–4.85	<0.01
	unknown	1.33	0.18–10.16	0.78
Blood transfusion	no	ref		
	yes	1.98	1.1–3.57	0.02
Neoadjuvant chemotherapy	no	ref		
	yes	0.9	0.48–1.7	0.75
AJCC system	0	ref		
	I	1.62	0.27–9.72	0.6
	II	2.21	0.55–8.85	0.26
	IIIa	4.11	1.25–13.5	0.02
	IIIb	12.92	3.61–46.27	<0.01
COBRA nomogram		1.46	1.22–1.74	<0.01
Simplified COBRA	low risk	ref		
	intermediate risk	2.27	1.19–4.33	0.01
	high risk	7.67	3–19.58	<0.01
ASA	2	ref		
	1	0.75	0.17–3.24	0.7
	3	1.17	0.64–2.14	0.61
	4	1.91	0.75–4.85	0.17
Caprini score	≤12	ref		
	>12	0.83	0.44–1.55	0.55
Improve score	≤3	ref		
	>3	1.17	0.58–2.36	0.65
Charlson comorbidity index		1.18	1.01–1.37	0.04
Non-age-adjusted Charlson Comorbidity Index		1.21	1.00–1.46	0.05
High Charlson comorbidity index	≤5	ref		
	>5	1.95	1.12–3.37	0.02
Neutrophil to lymphocyte ratio	≤3.5	ref		
	>3.5	1.87	1.09–3.23	0.02
Systemic immune-inflammation index	≤650	ref		
	>650	2.63	1.28–5.41	0.01
Neutrophil to erythrocyte ratio	≤1.2	ref		
	>1.2	2.035	1.07–3.88	0.03
Hemoglobin concentration	≥12 g/dL	ref		
	<12 g/dL	1.38	0.8–2.39	0.25

BMI—body mass index; CIS—carcinoma in situ; HG—high-grade; LG—low-grade; TCC—transitional cell carcinoma; SCC—squamous cell carcinoma; AJCC—American Joint Committee on Cancer; COBRA—Cancer of the Bladder Risk Assessment; ASA—American Society of Anesthesiology; HR—hazard ratio; CI—confidence intervals.

**Table 4 biomedicines-12-01541-t004:** Multivariable analyses with Cox proportional hazard for cancer-specific mortality among patients with bladder cancer who underwent radical cystectomy.

Multivariable Analysis for Cancer-Specific Mortality				
Variables		HR	95% CI	*p*-Value
AJCC system	0	ref		
	I	1.3	0.20–8.31	0.78
	II	2.59	0.62–10.74	0.19
	IIIa	3.39	1.01–11.39	0.05
	IIIb	23.25	6.06–89.26	<0.01
Charlson comorbidity index	≤5	ref		
	>5	2.72	1.45–5.11	<0.01
Smoking status	former	ref		
	never	2.71	1.06–6.9	0.04
	active	2.85	1.47–5.53	<0.01
	unknown	2.98	0.35–25.03	0.32
Concomitant CIS	no	ref		
	yes	18.09	3.41–96.03	<0.01
Histology	TCC	ref		
	other	21.24	3.97–113.73	<0.01
	SCC	2.31	0.27–20.18	0.45
Lymph node yield		0.91	0.86–0.97	<0.01
Neutrophil to lymphocyte ratio	>3.5	ref		
	≤3.5	1.80	1.01–3.21	0.046

TCC—transitional cell carcinoma; SCC—squamous cell carcinoma; CIS—carcinoma in situ; AJCC—American Joint Committee on Cancer; HR—hazard ratio; CI—confidence intervals.

**Table 5 biomedicines-12-01541-t005:** Multivariable analyses with Cox proportional hazard for overall mortality among patients with bladder cancer who underwent radical cystectomy.

Multivariable Analysis for Overall Mortality				
Variables		HR	95% CI	*p*-Value
AJCC system	0	ref		
	I	2.34	0.56–9.88	0.25
	II	1.79	0.49–6.56	0.38
	IIIa	3.02	1.06–8.63	0.04
	IIIb	18.34	5.47–61.51	<0.01
Charlson comorbidity index	≤5	ref		
	>5	1.82	1–3.3	0.05
Smoking status	former	ref		
	never	2.02	0.87–4.71	0.10
	active	2.55	1.36–4.78	<0.01
	unknown	2.1	0.26–17.08	0.49
Concomitant CIS	no	ref		
	yes	9.49	1.92–46.94	0.01
Histology	TCC	ref		
	other	10.84	2.13–55.13	0.01
	SCC	2.31	0.26–20.94	0.46
Lymph node yield		0.92	0.87–0.96	<0.01
Age		1.04	1–1.08	0.09
Systemic immune-inflammation index	≤650	ref		
	>650	1.99	1.03–3.87	0.04

TCC—transitional cell carcinoma; SCC—squamous cell carcinoma; CIS—carcinoma in situ; AJCC—American Joint Committee on Cancer; HR—hazard ratio; CI—confidence intervals.

## Data Availability

The original contributions presented in the study are included in the article, further inquiries can be directed to the corresponding author/s.

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
