# Peer review of "Can We Go beyond Pathology? The Prognostic Role of Risk Scoring Tools for Cancer-Specific Survival of Patients with Bladder Cancer Undergoing Radical Cystectomy"

_biomedicines, 2024, doi:10.3390/biomedicines12071541_

Round 1

Reviewer 1 Report

Comments and Suggestions for Authors

Authors have written a research article entitled “Can we go beyond pathology? The prognostic role of risk scoring tools for cancer-specific survival of patients with bladder cancer undergoing radical cystectomy”. The theme is interesting; however, the manuscript is not properly discussed in context to results. The data/experimental findings were not properly presented. Therefore, the manuscript is not suitable for publication; it should be carefully revised before the following peer-review process.

·       Authors first explain about the most recent and important achievements in the field related to present work. In my opinion, answers to these questions should be emphasized.

·       Authors have not explained the rationale of the study to a greater extent. How could this study enhance the understanding/knowledge of researcher or clinicians by using this type of data. Is this research could enhance the prognostic value of bladder cancer patients?

·       For better understanding of readers, authors must include a flow diagram representing the inclusion and exclusion criteria and methodology.

·       Results presented in this study are very limited, all the experimental analysis based upon secondary data of cancer. These data could not lead to the conclusion of present study. Some more parameters must be added in this study.

·       There is poor presentation of data like very less use of graph and figure.

·       The discussion should be rather organized around arguments avoiding simply describing details without providing much meaning. Authors have discussed some content and then seems not interested in writing more content which makes discussion part incomplete.

Comments on the Quality of English Language

Minor English corrections are required

Author Response

Authors have written a research article entitled “Can we go beyond pathology? The prognostic role of risk scoring tools for cancer-specific survival of patients with bladder cancer undergoing radical cystectomy”. The theme is interesting; however, the manuscript is not properly discussed in context to results. The data/experimental findings were not properly presented. Therefore, the manuscript is not suitable for publication; it should be carefully revised before the following peer-review process.

Author Response: We are very grateful for all the supportive comments and suggestions provided by the Reviewer. We attempted to introduce the required corrections and answer all of the posed questions. We hope that details and additional statistics provided within our answers will be considered sufficient. Modified and added sentences have been clearly marked in red within the manuscript.

  • Authors first explain about the most recent and important achievements in the field related to present work. In my opinion, answers to these questions should be emphasized.

Author Response: We are thankful for this thoughtful and kind suggestion. The additional paragraph has been added to the discussion section to present the most recent achievements in the field (lines 365-377).  This also gives context to our research and the rationale for using risk models when counseling patients undergoing radical cystectomy.

„The rationale for this study comes from the need for a better understanding of the aggressiveness of urothelial carcinoma which despite great advances in oncological and urological care is associated with poor prognosis. Many studies evidenced that within the last 20 years, the survival following RC did not improve significantly. This situation could have been attributed to the biological aggressiveness of the disease but also to the lack of effective systemic therapy or surgical inadequacy. Surgical waiting lists are another important factor known to affect survival due to very rapid disease progression implying a short window of opportunity for surgery with curative intent. No better treatment than RC has been invented neither for MIBC nor for BCG-unresponsive NMIBC despite the emergence of bladder-sparing approaches with trimodal therapy including radiation, chemotherapy, and maximal transurethral resection. However, some studies even suggested the equivocal outcomes of RC and TMT, but confounders and selection bias are the major limitations of these reports.”

  • Authors have not explained the rationale of the study to a greater extent. How could this study enhance the understanding/knowledge of researcher or clinicians by using this type of data. Is this research could enhance the prognostic value of bladder cancer patients?

Author Response: We are thankful for this thoughtful comment. An additional paragraph has been added to the discussion section to describe the rationale for the study and to better present the practical utility of results in clinical practice (lines 378-386).

“In this context, our study provides up-to-date outcomes of RC which are not satisfactory and identifies limitations of current scoring systems and risk tools. We believe that AJCC, COBRA and Charlson comorbidity index should be routinely used to risk-stratify patients and select candidates for adjuvant therapies (e.g. nivolumab/ radiation/ platin-based chemotherapy), but in the same time, we should seek for new biomarkers to personalize treatment. Most promising biomarkers include cell-free tumor DNA, which has been elegantly shown to predict the benefits of adjuvant therapy with atezolizumab. Meanwhile, our risk model provides easy-to-use prognostic factors that can be listed during postoperative counseling of patients to offer adjuvant therapy and present the risk.”

  • For better understanding of readers, authors must include a flow diagram representing the inclusion and exclusion criteria and methodology.

Author Response: We are thankful to the Reviewer for this notice. We appreciate your suggestion to clarify the inclusion and exclusion criteria using the figure. A flow diagram has been added to provide a clear view of selection criteria (figure 1) (page 3).

  • Results presented in this study are very limited, all the experimental analysis based upon secondary data of cancer. These data could not lead to the conclusion of present study. Some more parameters must be added in this study.

Author Response: Thank you for your inquiry. To improve the readability of results tables were re-formatted and two new tables were created. Additional parameters were added to the figure 1 which was modified and re-formatted. Namely, ASA, NLR and Caprini score were presented in figure 1 E-F. Additional baseline characteristics were listed in the results section. Additionally graphical visualization of multivariable model results were presented in the figure 3 (page 8).

  • There is poor presentation of data like very less use of graph and figure.

Author Response: Thank you for this relevant suggestion. Additional figure was added to present the flow chart and more Kaplan-Meier curves visualizing risk factors were presented in the figure 2 as listed above. Moreover, for the sake of better presentation of results table 1 and table 3 were split. Additional Figure 3 (page 12) was added to the manuscript as a graphical visualization of a multivariable risk model for CSM. We are thankful for this important comment and feel that our manuscript was improved after providing more figures and tables.

  • The discussion should be rather organized around arguments avoiding simply describing details without providing much meaning. Authors have discussed some content and then seems not interested in writing more content which makes discussion part incomplete.

Author Response: Thank you for this relevant suggestion. We acknowledge the importance of expanding the discussion section of our manuscript.

Discussion was improved with additional paragraphs describing current findings and challenges in bladder cancer, rationale of our study in the context of novel adjuvant treatment and more arguments for considering smoking status, concomitant CIS, histology type and neutrophil to lymphocyte ratio. Please see the additional sentences in lines 436-444.

“A previous study has demonstrated that quitting smoking more than 10 years before RC mitigates the risk of recurrence and CSM following the procedure. Furthermore, a lack of balance between immune competence and inflammatory response reflected by indices such as neutrophil-to-lymphocyte ratio is a well-recognized risk factor for adverse oncological outcomes following different urological surgeries, including RC. A meta-analysis of eighteen studies demonstrated that higher NLR was associated with recurrence-free and overall survival. Non-urothelial histology was shown as a risk factor associated with advanced stage and worse mortality. Concomitant CIS was suggested by Naspro et al. as a proxy of disease aggressiveness and portended worse survival outcomes.”

Reviewer 2 Report

Comments and Suggestions for Authors

The manuscript entitled “Can we go beyond pathology? The prognostic role of risk scoring tools for cancer-specific survival of patients with bladder cancer undergoing radical cystectomy
” was submitted to Biomedicines as research article for possible considerations. As known, radical cystectomy (RC) is a mainstay surgical treatment for non-metastatic muscle-invasive and BCG-unresponsive bladder cancer (BC).
In this study, the authors evaluated the prognostic role of various perioperative risk scores in BC patients undergoing RC, focusing on their ability to predict cancer-specific mortality (CSM). The study was nicely designed and analyzed. Authors well focused on their subject.

In this retrospective bicenter study, the authors evaluated the accuracy of different risk scores in the prediction of CSM following radical cystectomy for bladder cancer in search of greater personalization of the available clinical risk assessment tools. They concluded that the Low accuracy of AJCC staging system in the prediction of CSM, could be improved with the inclusively pathological data, Charlson comorbidity index, smoking history, and inflammatory indices, etc. The current report is helpful for refining risk assessment protocols and incorporating additional factors in CSM prognostication.

Minor issues:

1)     The analysis involved 215 patients who underwent RC between 2015 and 2021, however, the background of these patients was not depicted. So, if clinical doctors or researchers living in other countries and specific areas, want to compare the predication results, but the comparison may be restricted or inflected by the patients’ differences varying all the time based on genetic background and living environment, from races to customs, from resident places to climates, or from geo to diets.

2)     I guess authors may find way to put Table 1 into a single page. The present form of Table 1, occupied several pages. Why not cut the length of Table 1 in two halves, squeeze the current table to the left and set up two columns and put the second half to the right. Then the density of information is a little bit higher.

3)     As for Figure 1, three panels should be piled-up and enlarged. The current version was difficult to see the details in the Kaplan Meier curves illustrating cancer-specific survival. And please delete the box surrounding the panels. Make the figures clean and clear.

4)     Regarding the X-axis of Fig. 1, the unit of “survival” was day, month, or year? Or numbers of the patients? Readers may be confused. Hard to see the curves clearly.

5)     P<0.05 in the Fig.1, as you see, the first alphabet was not necessary to be capitalized. And the caption was incomplete. Fig.1 should be reformatted properly.

6)     Table 3 should be separated into two tables.

7)     How to cluster the data and visualized the results, are important. There was Only One figure in this manuscript.

8)     I agree with what authors declaration. But a graphical abstract is missing but kindly required to provide. A good GA can easily get people.

Minor revision.

Author Response

Reviewer 2

The manuscript entitled “Can we go beyond pathology? The prognostic role of risk scoring tools for cancer-specific survival of patients with bladder cancer undergoing radical cystectomy” was submitted to Biomedicines as research article for possible considerations. As known, radical cystectomy (RC) is a mainstay surgical treatment for non-metastatic muscle-invasive and BCG-unresponsive bladder cancer (BC). In this study, the authors evaluated the prognostic role of various perioperative risk scores in BC patients undergoing RC, focusing on their ability to predict cancer-specific mortality (CSM). The study was nicely designed and analyzed. Authors well focused on their subject.

Author Response: We are sincerely grateful for the supportive comments and insightful suggestions provided by the Reviewer. We have diligently incorporated the necessary corrections and addressed all of the posed questions. We trust that the corrections provided within our responses adequately address any concerns. Modified and added sentences have been clearly marked in red within the manuscript.

In this retrospective bicenter study, the authors evaluated the accuracy of different risk scores in the prediction of CSM following radical cystectomy for bladder cancer in search of greater personalization of the available clinical risk assessment tools. They concluded that the Low accuracy of AJCC staging system in the prediction of CSM, could be improved with the inclusively pathological data, Charlson comorbidity index, smoking history, and inflammatory indices, etc. The current report is helpful for refining risk assessment protocols and incorporating additional factors in CSM prognostication.

Author Response: Thank you for your thoughtful and encouraging feedback. We sincerely appreciate your recognition of the importance of our study in addressing current challenges in bladder cancer patients undergoing radical cystectomy.

Minor issues:

1)     The analysis involved 215 patients who underwent RC between 2015 and 2021, however, the background of these patients was not depicted. So, if clinical doctors or researchers living in other countries and specific areas, want to compare the predication results, but the comparison may be restricted or inflected by the patients’ differences varying all the time based on genetic background and living environment, from races to customs, from resident places to climates, or from geo to diets.

Author Response: Thank you for your insightful comments and suggestions. We acknowledge that this validation was performed in the cohort of patients from the central-eastern European origin, relatively homogenous population. Additional sentences were added to address that issue (lines 176-180):

In the results:

“All patients were of white race and Central-Eastern European origins and were treated in the academic institution. The vast majority of patients were active or former smokers (178; 83%) and only 26 (12%) denied smoking. Median BMI was 25.3 kg/m² with interquartile range between 22.2 and 29.3 kg/m².”

In the discussion (lines 447-448):

“The results of our study and validation analysis must be contextualized within the relatively homogeneous Central-Eastern European origins of the included patients.”

2)     I guess authors may find way to put Table 1 into a single page. The present form of Table 1, occupied several pages. Why not cut the length of Table 1 in two halves, squeeze the current table to the left and set up two columns and put the second half to the right. Then the density of information is a little bit higher.

Author Response: Thank you for your thoughtful comments to improve our table. Please see corrected table 1 which was split to two tables.

3)     As for Figure 1, three panels should be piled-up and enlarged. The current version was difficult to see the details in the Kaplan Meier curves illustrating cancer-specific survival. And please delete the box surrounding the panels. Make the figures clean and clear.

Author Response: Thank you for your thoughtful suggestions to improve our figure. Please see corrected figure 1 (re-named as figure 2 currently) with additional Kaplan-Meier curves which were all enlarged and re-formatted.

4)     Regarding the X-axis of Fig. 1, the unit of “survival” was day, month, or year? Or numbers of the patients? Readers may be confused. Hard to see the curves clearly.

Author Response: Thank you for your thoughtful suggestions to improve our figure. Please see corrected figure 1 (re-named as figure 2 currently) with additional Kaplan-Meier curves and added description below the figure.

“Figure 2. Kaplan Meier curves with confidence intervals illustrating cancer-specific survival (months) according to (A) AJCC system, (B) simplified COBRA stratification and (C) Charlson Comorbidity Index (CCI >5 vs ≤ 5), (D) ASA scale, (E) Caprini score (>6 vs ≤ 6) and (F) neutrophil to lymphocyte ratio (>3.5 vs ≤ 3.5).

Numbers at risk are presented below the curves at specific time points during follow-up measured within months following RC.”

5)     P<0.05 in the Fig.1, as you see, the first alphabet was not necessary to be capitalized. And the caption was incomplete. Fig.1 should be reformatted properly.

Author Response: Thank you for your thoughtful suggestions to improve our figure. P-value was corrected.

6)     Table 3 should be separated into two tables.

Author Response: Thank you, following your suggestion we separated table 3 into two tables – listed as 4 and 5 after building extra tables as described above.

7)     How to cluster the data and visualized the results, are important. There was Only One figure in this manuscript.

Author Response: Thank you for your important input. Additional figures were added including graphical visualization of selection criteria and presentation of more data within Kaplan-Meier curves and graphical presentation of main findings regarding risk factors for CSM. Please see figure 1 and 2 (A-F) and 3 within the manuscript. Tables were also multiplied to present the information more clearly and limited to one page per table. Thank you for all your supportive comments. We believe that our manuscript has been improved.

8)     I agree with what authors declaration. But a graphical abstract is missing but kindly required to provide. A good GA can easily get people.

Author Response: Thank you for your important suggestion. Figure 3 (page 12) was added to the manuscript as a graphical visualization of a multivariable risk model for CSM.

Reviewer 3 Report

Comments and Suggestions for Authors

The common perioperative scoring tools of comorbidity burden were evaluated ,n a retrospective study for 215 radical cystectomy treated bladder cancer patients. AJCC staging system and COBRA risk score were found to be low value in predicting. But, AJCC system can be improved with the inclusion additional pathological data, lymph node counts, Charlson comorbidity index, smoking history and systemic inflammatory marker. The manuscript is well written and based on analysis as presented. I think the results might be useful for further studies to relate bladder cancer prediction with appropriate scoring tools.

Author Response

Author Response: Thank you for your thoughtful and encouraging feedback. We sincerely appreciate your recognition of the importance of our study in addressing current challenges in bladder cancer patients undergoing radical cystectomy.

Round 2

Reviewer 1 Report

Comments and Suggestions for Authors

Authors have significantly improved the current version of the manuscript. Hence, I have no objection in consideration of the manuscript however authors are suggested to include some latest clinical studies in the introduction and discussion section:

https://doi.org/10.1016/j.clgc.2023.06.011

https://doi.org/10.1155/2024/3885057

https://doi.org/10.1155/2024/1372188

https://doi.org/10.56434/j.arch.esp.urol.20227510.127

Additionally manuscript should be carefully proofread for grammatical and typos errors.

Comments on the Quality of English Language

Minor editing of English language required

Author Response

Review

Authors have significantly improved the current version of the manuscript. Hence, I have no objection in consideration of the manuscript however authors are suggested to include some latest clinical studies in the introduction and discussion section:

https://doi.org/10.1016/j.clgc.2023.06.011

https://doi.org/10.1155/2024/3885057

https://doi.org/10.1155/2024/1372188

https://doi.org/10.56434/j.arch.esp.urol.20227510.127

Additionally manuscript should be carefully proofread for grammatical and typos errors.

Our response

Thank you for your positive feedback and constructive suggestions. We are pleased to hear that the current version of the manuscript has significantly improved. We incorporated the latest clinical studies including the study that you recommended into the manuscript (https://doi.org/10.1016/j.clgc.2023.06.011; DOI: 10.3390/ijms24065846).

Additionally, we carefully proofread the manuscript to address any grammatical and typographical errors. Thank you once again for your valuable input. A corrected version of the manuscript with changes highlighted in red was resubmitted.